# Multimorbidity Patterns and Unplanned Hospitalisation in a Cohort of Older Adults

**DOI:** 10.3390/jcm9124001

**Published:** 2020-12-10

**Authors:** Roselyne Akugizibwe, Amaia Calderón-Larrañaga, Albert Roso-Llorach, Graziano Onder, Alessandra Marengoni, Alberto Zucchelli, Debora Rizzuto, Davide L. Vetrano

**Affiliations:** 1Aging Research Center, Department of Neurobiology, Care Sciences and Society, Karolinska Institutet and Stockholm University, 17165 Solna, Sweden; alessandra.marengoni@unibs.it (A.M.); debora.rizzuto@ki.se (D.R.); davide.vetrano@ki.se (D.L.V.); 2Fundació Institut Universitari per a la Recerca a l’Atenció Primària de Salut Jordi Gol i Gurina (IDIAPJGol), 08007 Barcelona, Spain; aroso@idiapjgol.org; 3Campus de la UAB, Universitat Autònoma de Barcelona, 08193 Bellaterra (Cerdanyola del Vallès), Spain; 4Department of Cardiovascular, Endocrine-Metabolic Diseases and Aging, Istituto Superiore di Sanità, 00161 Rome, Italy; graziano.onder@iss.it; 5Department of Clinical and Experimental Sciences, University of Brescia, 25123 Brescia, Italy; 6Department of Information Engineering, University of Brescia, 25123 Brescia, Italy; a.zucchelli001@unibs.it; 7Stockholm Gerontology Research Centrum, 11346 Stockholm, Sweden; 8Centro Medicina dell’Invecchiamento, Fondazione Policlinico Universitario “A. Gemelli” IRCCS, and Università Cattolica del Sacro Cuore, 00168 Rome, Italy

**Keywords:** hospitalisation, multimorbidity, older adults, person-centred care

## Abstract

The presence of multiple chronic conditions (i.e., multimorbidity) increases the risk of hospitalisation in older adults. We aimed to examine the association between different multimorbidity patterns and unplanned hospitalisations over 5 years. To that end, 2,250 community-dwelling individuals aged 60 years and older from the Swedish National Study on Aging and Care in Kungsholmen (SNAC-K) were studied. Participants were grouped into six multimorbidity patterns using a fuzzy c-means cluster analysis. The associations between patterns and outcomes were tested using Cox models and negative binomial models. After 5 years, 937 (41.6%) participants experienced at least one unplanned hospitalisation. Compared to participants in the *unspecific* multimorbidity pattern, those in the *cardiovascular diseases, anaemia and dementia* pattern, the *psychiatric disorders* pattern and the *metabolic and sleep disorders* pattern presented with a higher hazard of first unplanned hospitalisation (hazard ratio range: 1.49–2.05; *p* < 0.05 for all), number of unplanned hospitalisations (incidence rate ratio (IRR) range: 1.89–2.44; *p* < 0.05 for all), in-hospital days (IRR range: 1.91–3.61; *p* < 0.05 for all), and 30-day unplanned readmissions (IRR range: 2.94–3.65; *p* < 0.05 for all). Different multimorbidity patterns displayed a differential association with unplanned hospital care utilisation. These findings call for a careful primary care follow-up of older adults with complex multimorbidity patterns.

## 1. Background

Multimorbidity, the co-occurrence of two or more chronic conditions in an individual [1,2,3,4], is expanding globally due to people’s longer life expectancy [5,6,7]. The prevalence of multimorbidity is highest among the older population [8], ranging from 38–76% in persons aged 60–74 years to 76–88% among persons aged 85 years or older [9,10].

The consequences of multimorbidity affect both individuals and health care systems [4], since multimorbidity has been associated with poorer quality of life and mental health, impaired functional ability, and increased health care costs [11,12,13]. Such elevated health care costs are attributed, among other factors, to an increased utilisation of primary care and, especially, hospital care [11,12]. A European-wide study carried out in the general population among participants aged 50 years or older showed that multimorbidity increases the likelihood of being hospitalised by 49%, the number of hospital admissions by 35%, and the length of hospital stay by 49%, compared to participants without multimorbidity [13]. Another retrospective study carried out in three developed countries involving patients with multimorbidity discharged from medical inpatient wards revealed that 15.6% of them were readmitted within 30 days from discharge and 9.6% of these readmissions were potentially avoidable [14].

Most previous studies on multimorbidity and hospital care use have assessed multimorbidity as the mere count of chronic diseases [11,15,16], but previous research has shown that chronic diseases tend to cluster together into so-called multimorbidity patterns [17,18,19]. Despite the varying study populations and methodologies, three patterns of multimorbidity involving cardiometabolic diseases, psychogeriatric problems and mechanical and somatoform disorders have been consistently suggested to be very prevalent in the older population [20]. The few studies that have looked at multimorbidity patterns and hospital care utilisation have some limitations, such as including only subjects aged 80 years or older [21,22] or already-hospitalised patients [23,24].

The aim of the present study was to examine the association between multimorbidity patterns and 5-year unplanned hospital care utilisation in community-dwelling older adults, in terms of time to first hospitalisation, and the cumulative number of hospitalisations, in-hospital days and 30-day readmissions.

## 2. Methods

### 2.1. Study Population

This was a prospective cohort based on data from the Swedish National study on Aging and Care in Kungsholmen (SNAC-K; https://www.snac-k.se/). SNAC-K is an ongoing longitudinal study that included a random sample from 11 age cohorts of people aged 60 years or older (range 60–104) at baseline [25]. The 11 age cohorts were 60, 66, 72, 78, 81, 84, 87, 90, 93, 96 and 99 years or older. At baseline (years 2001–2004), the participation rate was 73.3%, leading to a study population of 3,363 individuals living either in the community or in institutions in the Kungsholmen central area of Stockholm, Sweden.

The present study included only participants with multimorbidity, therefore, 439 participants with less than 2 chronic diseases were excluded. In addition, 183 participants living in institutions were excluded, considering that they received medical and nursing care that might strongly modify the profiles of hospital care use [26]. Hence, only participants living in the community at baseline were included. We also excluded 491 additional participants hospitalised 1 year before their entry in the study in light of the evidence showing that hospitalisation is a major risk factor for future rehospitalisation [14]. After the exclusions, the final analytical sample consisted of 2250 participants. Approval for the study was obtained from the Regional Ethics Review Board in Stockholm County. Participants signed informed consent forms or a proxy was contacted if the participant was cognitively impaired.

### 2.2. Chronic Disease Assessment

Diagnoses were performed based on participants’ medical history, clinical examinations, self-reported information and proxy interviews. Clinical parameters, laboratory tests, inpatient and outpatient care data and medications were used to complement diagnostic data according to a methodology previously proposed by our group [10]. Chronic diseases were coded in accordance with the International Classification of Diseases, 10th revision (ICD10) and were subsequently grouped into 60 homogeneous categories (Appendix A).

### 2.3. Outcomes Definition

Participants’ hospital care utilisation was derived from the Swedish National Patient Register. The following information was collected from the register for the first 5 years since the baseline SNAC-K assessment: (i) first ever unplanned hospitalisation; (ii) number of unplanned hospitalisations; (iii) number of in-hospital days due to unplanned admissions; (iv) number of unplanned readmissions within 30 days of discharge.

### 2.4. Covariates

Information on age, sex, education (elementary, high school and university or above), marital status (unmarried, married, divorced and widowed), living arrangement, alcohol use (never/occasional, light/moderate and heavy), and smoking habits (never, former and current) was gathered through nurse-led questionnaires. Height and weight were measured and participants’ body mass index (BMI; kg/m^2^) was computed.

### 2.5. Analytical Approach

Multimorbidity patterns were identified following a procedure described previously by our research group [27]. Briefly, a fuzzy c-means cluster analysis was employed to classify study participants into clusters of systematically coexisting chronic diseases and a membership matrix was obtained with each subject’s probability of belonging to the different multimorbidity patterns. Diseases were deemed to characterise a given cluster if they displayed an observed-to-expected (O/E) ratio ≥2 and an exclusivity ≥25% [27,28]. O/E ratios were calculated by dividing the prevalence of a given disease within a cluster by its prevalence in the study population. Disease exclusivity refers to the fraction of participants with the disease included in the cluster over the total number of participants with the disease.

The association between multimorbidity patterns and time to the first unplanned hospitalisation was tested though Cox proportional hazard regression models (hazard ratios (HR) and 95% confidence intervals (CI)), upon verifying the proportional hazards assumption using the global test and Kaplan–Meier curves. Models were censored for death, which was derived from the Swedish Death Register. All outcomes in the present study were over-dispersed due to the large number of participants who did not experience them. Thus, negative binomial regression models were used to test the association between multimorbidity patterns and the number of hospitalisations, in-hospital days, and unplanned readmissions within 30 days from discharge (incidence rate ratio (IRR) and 95% CI). For the outcome “in-hospital days,” participants who were not hospitalised were considered to have 0 days of hospitalisation. The exposure time, which was 5 years or the time between baseline and the date of death if shorter than 5 years, was accounted for using person-years as the offset in the models. The time spent in the hospital over the 5 years was subtracted from the exposure time. All models were adjusted by age, sex and education level, and additionally by alcohol consumption and smoking. Marital status was also included as a covariate in the models, but given its lack of confounding effect, it was finally omitted. The *unspecific* pattern was employed as the reference group, given that subjects belonging to this pattern had the lowest number of chronic diseases, disabilities and mortality risk [27,28]. Interactions between multimorbidity patterns with age and sex were tested and stratified analyses were carried out. Analyses were performed using Stata IC/15.0 and R version 4.0.0. A *p*-value of <0.05 was considered to be statistically significant.

## 3. Results

Six patterns of multimorbidity were identified (Appendix A): (i) *Psychiatric disorders*, including 5.87% of participants; (ii) *Cardiovascular diseases, anaemia and dementia*, including 6.27% of participants; (iii) *Metabolic and sleep disorders*, including 10.67% of participants; [iv] *Sensory impairments and cancer*, including 11.87% of participants; (v) *Musculoskeletal, respiratory and gastrointestinal diseases* including 15.78% of participants; and (vi) *Unspecific*, including 49.56% of participants. The unspecific pattern was different from the other patterns because none of the diseases included presented both an exclusivity ≥ 25% and an O/E ratio ≥ 2. The baseline characteristics of study participants across multimorbidity patterns are shown in Table 1.

After 5 years of follow-up, 394 participants died, mostly in the *cardio/anaemia/dementia* pattern (51.8%) and the *sensory/cancer* pattern (35.2%) (Appendix A). At the end of the follow-up, 937 (41.6%) participants experienced at least one unplanned hospitalisation. Subjects in the *cardio/anaemia/dementia* pattern (HR 2.05, 95% CI 1.62, 2.61), the *metabolic/sleep disorders* pattern (HR 1.50, 95% CI 1.20, 1.86) and the *psychiatric disorders* pattern (HR 1.49, 95% CI 1.13, 1.96) showed a significantly higher hazard of a first unplanned hospitalisation, compared to the unspecific pattern. Participants in the *sensory/cancer pattern* also showed a higher hazard of unplanned hospitalisation (HR 1.24, 95% CI 1.01, 1.52), compared to the *unspecific* pattern. However, the statistical significance became borderline in the fully adjusted model (Table 2, Figure 1).

During the follow-up, participants in the *cardio/anaemia/dementia* pattern experienced the highest number of unplanned hospitalisations, 4.73 per 10 person-years. In multivariate analyses, subjects belonging to the *cardio/anaemia/dementia* pattern (IRR 2.44, 95% CI 1.82, 3.28), the *metabolic/sleep disorders* pattern (IRR 1.93, 95% CI 1.52, 2.45), and the *psychiatric disorders* pattern (IRR 1.89, 95% CI 1.37, 2.59) presented with a significantly higher risk of unplanned hospitalisation, compared to subjects in the *unspecific* pattern. Although participants in the *sensory/cancer* pattern also showed a higher risk of unplanned hospitalisation compared to subjects in the *unspecific* pattern (HR 1.31, 95% CI 1.03, 1.65), the statistical significance became borderline in the fully adjusted model (Table 2).

Participants in the *psychiatric disorders* pattern were found to have the longest hospital stay, 122.72 days per 10 person-years. Indeed, the risk of longer in-hospital stay was highest for the *psychiatric disorders* pattern (IRR 3.61, 95% CI 2.16, 6.04), followed by the *cardio/anaemia/dementia* pattern (IRR 2.07, 95% CI 1.23, 3.49) and the *metabolic/sleep disorders* pattern (IRR 1.91, 95% CI 1.27, 2.87), compared to the *unspecific* pattern (Table 2).

Subjects in the *metabolic/sleep disorders* pattern presented with the highest incidence rate of unplanned 30-day readmissions (0.99 per 10 person-years) and had the highest risk of readmission in multivariate analyses (IRR 3.65, 95% CI 2.16, 6.14), followed by the *psychiatric disorders* pattern (IRR 3.00, 95% CI 1.44, 6.28) and the *cardio/anaemia/dementia* pattern (IRR 2.94, 95% CI 1.55, 5.56), compared to the *unspecific* pattern (Table 2).

No statistically significant interactions were detected between multimorbidity patterns and age and sex. In age-stratified analyses, the associations with the *psychiatric disorders* pattern and the *metabolic/sleep disorders* pattern were strengthened among young-old participants, while the opposite was true for the *cardio/anaemia/dementia* pattern (Appendix A).

## 4. Discussion

This study investigated the association between different disease patterns and unplanned hospital care utilisation in multimorbid older adults. The findings of this study showed that older individuals belonging to specific multimorbidity patterns had different risk of unplanned hospitalisation over a period of 5 years.

Belonging to the *cardiovascular diseases, anaemia and dementia* pattern and the *metabolic and sleep disorders* pattern was strongly associated with a higher unplanned hospital care utilisation. These two patterns were characterised by a high prevalence of cerebrovascular diseases, anaemia, ischemic heart disease, cardiac valve diseases, dementia, diabetes and obesity. An Italian study carried out among individuals aged 65 years and older using latent class analysis showed an increased risk of hospital admissions in people with heart diseases as well as with metabolic/ischemic patterns [23]. However, the associations were even stronger in our study, most likely due to differences in how the exposure was defined and measured and in the study population. Individuals with these disease patterns are more prone to experience organ decompensations and medical emergencies, which may explain the observed higher utilisation of unplanned hospital care [29,30,31]. In addition, these individuals may be at higher risk of coexisting conditions like infections, emotional or stressful events, poor medication adherence and the use of potentially harmful medications, which could further increase their risk of being hospitalised [32].

The *psychiatric disorders* pattern also showed a relatively strong association with unplanned hospital care utilisation, compared to the *unspecific* pattern. More specifically, individuals in this pattern had the highest risk of longer in-hospital stay and the second highest risk of 30-day readmissions. Previous evidence has shown that the presence of psychiatric conditions among individuals with physical multimorbidity increases the risk of unplanned hospitalisation and readmissions [33,34]. In addition, depression, which is a major constituent of this pattern, is associated with increased risk of emergency hospital admissions [35]. The findings of our study are consistent with a recent Danish study, which revealed that the neuropsychiatric and mental disorders patterns had higher odds of experiencing acute hospital admissions [24]. Similarly, another Italian study showed that multimorbidity dyads including depression had a higher risk of increased annual hospitalisation days [23]. These findings stress the need for a proper management of mental illness among multimorbid patients in order to reduce hospital care use and, consequentially, increase quality of life.

The *musculoskeletal, respiratory and gastrointestinal diseases* pattern showed a higher hazard of unplanned hospitalisation, but no association arose with other outcomes. Diseases like asthma and chronic obstructive pulmonary disease, which comprise this pattern, are characterised by frequent exacerbations, which are most likely responsible for the higher hazard of first unplanned hospitalisation [29]. These results are somewhat contrary to the findings from a study carried out in New Zealand, which showed a non-significant risk of hospitalisation for the pattern comprised by asthma, chronic obstructive pulmonary disease, osteoarthritis and peripheral vascular disease [22]. This discrepancy could be explained by the age differences between the two study groups. Whereas our study was carried out among older adults aged 60 years or older, the study from New Zealand was done among participants aged 80 to 90 years.

On borderline statistical significance, subjects in the *sensory/cancer pattern* showed a higher hazard of first hospitalisation and a higher number of hospitalisations, compared to those in the *unspecific* pattern. This is arguably due to cancer, which, in its severe and advanced forms, causes pain, psychological distress, complications of treatment and hospital emergency visits [36,37,38]. This is in line with an Italian study showing a 44% higher risk of hospitalisation among participants in the cancer pattern [23]. Subjects in this pattern could also be frailer, as indicated by their older age and higher prevalence of underweight, a typical feature of the frailty phenotype [39].

### Strengths and Limitations

We used data from a prospective population-based study (SNAC-K), in which participants’ clinical assessment was carried out in a very comprehensive way using multiple data sources, which reduces the risk of information bias. The study had a large sample size, of which many participants were hospitalised during the follow-up period providing sufficient statistical power. In addition, we used the Swedish National Patient Register to retrieve information on hospitalisation episodes, which may have limited the probability of having losses to follow-up or outcome misclassification. Unlike previous studies looking at multimorbidity patterns, we classified individuals (and not diseases) based on their underlying disease patterns, which better responds to a person-centred approach.

The results of the study should, however, be interpreted considering some limitations. The main exposure (i.e., multimorbidity patterns) was only measured at baseline. Thus, any changes in the exposure status could have diluted the studied associations, especially given the choice of using the *unspecific* pattern as the reference category. However, the prediction of future outcomes based on baseline clinical profiles better reflects real clinical practice, where medical prognostication is based on current health status. In addition, given that our study introduces a new way of conceptualising multimorbidity, comparison with previous literature is challenging. The generalisability of the study findings could have been affected by the fact that the majority of the study participants in SNAC-K are well educated and financially stable. Furthermore, we excluded individuals who had been hospitalised 1 year prior to the baseline assessment and those who at baseline were living in institutions. Exclusion of these individuals may have further limited the generalizability of our findings and weakened the strength of the associations, particularly for those patterns associated with higher unplanned hospital care utilisation.

These findings call for a careful primary care follow-up of those older adults with complex multimorbidity patterns, such as those with cardiovascular diseases, anaemia and dementia, so that they are prevented from a high unplanned hospital care utilisation. In older patients, hospitalisation is often the starting point of the so-called hospital-associated deconditioning, which is induced by a combination of physical inactivity, the underlying illness and inadequate disease management and its effects [40]. It affects a range of body systems and leads to disabilities, readmissions, increased demands for home care, institutionalisation, and mortality at discharge, particularly among older adults who are already deteriorated [41,42,43,44]. The findings of the present study can be also useful to clinicians to better assess the prognosis of multimorbid patients and counsel patients and their families accordingly. Indeed, the cluster approach to the operationalisation and study of multimorbidity is increasingly recognised as an optimal way to decipher the clinical heterogeneity and endless number of chronic disease combinations older individuals tend to suffer from.

## 5. Conclusions

The present study provides novel insights into unplanned hospital care utilisation by older individuals presenting with different multimorbidity patterns. It shows that different groups of individuals with varying multimorbidity patterns show different risk and intensity of unplanned hospital care utilisation. In particular, older adults displaying disease patterns characterised by cardiovascular disease, anaemia and dementia; metabolic and sleep disorders; and psychiatric disorders are responsible for the highest bulk of unplanned hospital care use. The results of this study may inform both clinicians and policymakers to better monitor and care for multimorbid older adults.

## Figures and Tables

**Figure 1 jcm-09-04001-f001:**
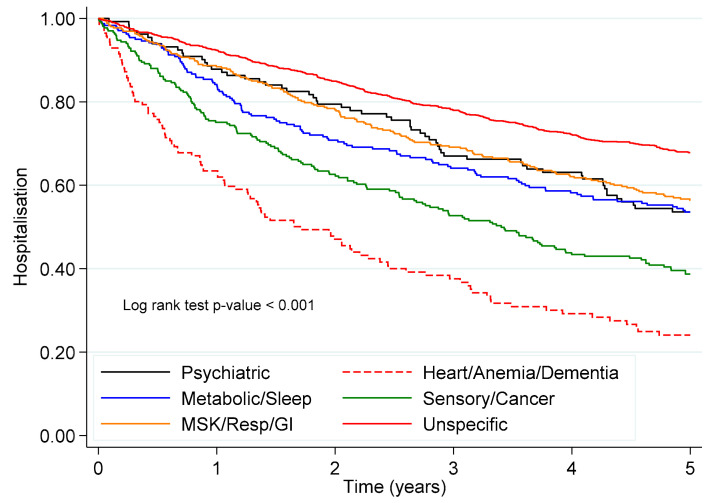
Kaplan–Meier survival estimates for the first unplanned hospitalisation throughout 5 years of follow-up. Abbreviations: GI—gastrointestinal diseases; Resp—respiratory diseases; MSK—musculoskeletal diseases; Cardio—cardiovascular diseases.

**Table 1 jcm-09-04001-t001:** Baseline characteristics of study participants across multimorbidity patterns.

	Psychiatric *n* = 132 (5.87%)	Cardio/Anaemia/Dementia *n* = 141 (6.27%)	Metabolic/Sleep *n* = 240 (10.67%)	Sensory/Cancer *n* = 267 (11.87%)	MSK/Resp/GI *n* = 355 (15.78%)	Unspecific *n* = 1115 (49.56%)
**Age, *n* (%)**						
< 78 years	80 (60.6)	20 (14.2)	138 (57.5)	43 (16.1)	174 (49.0)	744 (66.7)
≥ 78 years	52 (39.4)	121 (85.8)	102 (42.5)	224 (83.9)	181 (51.0)	371 (33.3)
**Sex, *n* (%)**						
Male	31 (23.5)	49 (34.8)	122 (50.8)	86 (32.2)	80 (22.5)	405 (36.3)
Female	101 (76.5)	92 (65.2)	118 (49.2)	181 (67.8)	275 (77.5)	710 (63.7)
**Education, *n* (%)**						
Elementary	23 (17.4)	34 (25.2)	37 (15.4)	62 (23.3)	59 (16.7)	170 (15.3)
High school	60 (45.5)	78 (57.8)	130 (54.2)	143 (53.8)	182 (51.4)	536 (48.2)
University	49 (37.1)	23 (17.0)	73 (30.4)	61 (22.9)	113 (31.9)	406 (36.5)
**Marital status, *n* (%)**						
Unmarried	24 (18.2)	23 (16.7)	34 (14.2)	46 (17.3)	69 (19.5)	188 (16.9)
Married	43 (32.6)	43 (31.2)	116 (48.5)	69 (25.9)	138 (39.0)	556 (50.0)
Divorced	31 (23.4)	11 (8.0)	31 (13.0)	34 (12.8)	52 (14.7)	145 (13.0)
Widowed	34 (25.8)	61 (44.1)	58 (24.3)	117 (44.0)	95 (26.8)	223 (20.1)
**Alcohol intake, *n* (%)**						
Never/occasional	46 (35.2)	73 (54.9)	87 (36.7)	142 (54.6)	139 (39.6)	324 (29.2)
Light/moderate	48 (36.6)	46 (34.6)	114 (48.1)	85 (32.7)	152 (43.3)	595 (53.7)
Heavy	37 (28.2)	14 (10.5)	36 (15.2)	33 (12.7)	60 (17.1)	190 (17.1)
**Smoking, *n* (%)**						
Never	53 (40.2)	73 (53.7)	97 (40.9)	147 (55.7)	175 (49.8)	516 (46.6)
Former	49 (37.1)	49 (36.0)	108 (45.6)	89 (33.7)	128 (36.5)	411 (37.2)
Current	30 (22.7)	14 (10.3)	32 (13.5)	28 (10.6)	48 (13.7)	179 (16.2)
**BMI, *n* (%)**						
Underweight	2 (1.6)	3 (2.7)	2 (0.9)	11 (4.9)	12 (3.5)	6 (0.6)
Normal weight	68 (56.2)	67 (59.3)	60 (25.8)	143 (63.3)	204 (60.0)	594 (54.8)
Overweight	44 (36.4)	33 (29.2)	80 (34.3)	67 (29.6)	104 (30.6)	394 (36.3)
Obese	7 (5.8)	10 (8.8)	91 (39.0)	5 (2.2)	20 (5.9)	90 (8.3)
**Chronic diseases, *n* (%)**						
≥3	123 (93.2)	140 (99.3)	227 (94.6)	259 (97.0)	314 (88.5)	676 (60.6)
≥4	111 (84.1)	136 (96.5)	195 (81.3)	239 (89.5)	243 (68.5)	287 (25.7)

Abbreviations: GI—gastrointestinal diseases; Resp—respiratory diseases; MSK—musculoskeletal diseases; Cardio—cardiovascular diseases; BMI—body mass index. Missing data: education 7, smoking 25, alcohol consumption 30, BMI 134.

**Table 2 jcm-09-04001-t002:** Association of multimorbidity patterns with unplanned hospital care utilisation.

**Multimorbidity Patterns**	**IR per 10 Person-Years**	**Model 1** **HR (95% CI)**	**Model 2** **HR (95% CI)**
**First hospitalisation**			
Unspecific	0.78	Ref	Ref
Psychiatric	1.23	1.53 (1.16, 2.01) **	1.49 (1.13, 1.96) **
Cardio/Anaemia/Dementia	3.36	2.11 (1.67, 2.67) ***	2.05 (1.62, 2.61) ***
Metabolic/Sleep	1.32	1.53 (1.23, 1.89) ***	1.50 (1.20, 1.86) ***
Sensory/Cancer	2.03	1.24 (1.01, 1.52) *	1.21 (0.99, 1.48)
MSK/Resp/GI	1.17	1.23 (1.01, 1.49) *	1.23 (1.01, 1.50) *
**Multimorbidity Patterns**	**IR per 10 Person-Years**	**Model 1** **IRR (95% CI)**	**Model 2** **IRR (95% CI)**
**Number of unplanned hospitalisations**			
Unspecific	1.94	Ref	Ref
Psychiatric	3.65	1.91 (1.39, 2.61) ***	1.89 (1.37, 2.59) ***
Cardio/Anaemia/Dementia	4.73	2.43 (1.82, 3.26) ***	2.44 (1.82, 3.28) ***
Metabolic/Sleep	3.74	1.96 (1.55, 2.48) ***	1.93 (1.52, 2.45) ***
Sensory/Cancer	2.42	1.31 (1.03, 1.65) *	1.25 (0.99, 1.58)
MSK/Resp/GI	2.39	1.23 (0.99, 1.52)	1.23 (1.00, 1.53)
**In-hospital days**			
Unspecific	33.95	Ref	Ref
Psychiatric	122.72	3.42 (2.07, 5.67) ***	3.61 (2.16, 6.04) ***
Cardio/Anaemia/Dementia	70.33	2.07 (1.23, 3.48) **	2.07 (1.23, 3.49) **
Metabolic/Sleep	64.95	1.99 (1.34, 2.98) ***	1.91 (1.27, 2.87) **
Sensory/Cancer	36.20	1.18 (0.79, 1.77)	1.07 (0.71, 1.60)
MSK/Resp/GI	40.20	1.16 (0.83, 1.62)	1.18 (0.84, 1.66)
**30-day readmissions**			
Unspecific	0.27	Ref	Ref
Psychiatric	0.82	3.13 (1.50, 6.51) **	3.00 (1.44, 6.28) **
Cardio/Anaemia/Dementia	0.80	2.82 (1.49, 5.35) **	2.94 (1.55, 5.56) ***
Metabolic/Sleep	0.99	3.66 (2.18, 6.16) ***	3.65 (2.16, 6.14) ***
Sensory/Cancer	0.41	1.65 (0.97, 2.80)	1.49 (0.87, 2.55)
MSK/Resp/GI	0.42	1.54 (0.94, 2.52)	1.54 (0.94, 2.52)

*p*-values: * = < 0.05, ** = < 0.01, *** = < 0.001. Model 1 adjusted by age, sex, education level. Model 2 adjusted by age, sex, education level, alcohol consumption, smoking. Abbreviations: GI—gastrointestinal diseases; Resp—respiratory diseases; MSK—musculoskeletal diseases; Cardio—cardiovascular diseases; HR—hazard ratio; CI—confidence interval; IR—incidence rate; IRR—incidence rate ratio.

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
