# Peer review of "Multimorbidity Patterns and Unplanned Hospitalisation in a Cohort of Older Adults"

_jcm, 2020, doi:10.3390/jcm9124001_

Round 1
Reviewer 1 Report
This appears to be an excellent study that generates insights into hospital care use by individuals with multiple chronic conditions. I found the manuscript very clear and I enjoyed reading it. As such, I think that this article is worthy of publication in JCM and my comments involve only minor suggestions (many of which are stylistic, and can therefore be ignored if the authors choose to do this).
Comments:
P.2, Lines 45-46. “The excess expenditure on patients with multimorbidity is estimated to be 2.6 million Euros per year”. Could the authors perhaps clarify for a general reader exactly what this means? Does this mean that 2.6 million euros extra is spent per year compared to if those patients only had a single chronic condition? Or compared to if they had no condition at all? Or compared to if those multiple conditions were instead split between different patients? The same question applies in the following sentence too.
P.2, Line 59. “80+ years”. This reads quite informally, and so I would amend to “80 years or older”. I think this is ok in the Methods, where a category name is being stated. (I would also check this throughout the manuscript).
The study population is 3363, and n=491 were excluded. Can the authors please explain where their figure of 2250 for the final sample figure comes from? Also, I’m not sure the “n” notation in “n=491” is useful here, since it is not used elsewhere – why not simply say that 491 participants were excluded?
Why did the authors choose an O/E ratio threshold of 2 and an exclusivity threshold of 25%? Is this a standard choice? Could the authors justify this, and also demonstrate the robustness of their results to these choices by showing their results for other cluster definitions (maybe as a supplementary analysis?)
Sorry to be slightly pedantic, but should “widow” be “widowed” in Table 1 for consistency with the other descriptors? Similarly, “high school”, “never/occasional” and “light/moderate” should probably be capitalised.
P.6, line 36 – “Metabolic/ sleep disorders” should be “Metabolic/sleep disorders” for consistency (sim. In figure 1, “P- value” should be “p-value” for consistency)
In figure 1, I found the line colours quite hard to differentiate (particularly psychiatric vs metabolic vs MSK) – could the authors perhaps choose colours that are more obviously different?
P.6 line 47. The same comment as above re “65+”. I would also change “done” to “carried out” for the same reason.
P 7 line 71 – is “conform” a typo? Maybe “conform to” or “confirm”?
In the Discussion, the authors do not appear to mention the sensory/cancer group. However this group, along with the cardio group, is most different to the unspecific group in Fig 1. Should this be discussed?
A couple of additional questions that I had as I was reading through are already addressed in the Strengths/Limitations section, so thanks for laying this out so clearly. (I would consider removing the “Furthermore… Furthermore” in lines 95-97 of page 7).
Finally, the references require thorough checking (e.g. the capitalisation of references 5 and 28 – which I assume was not the intention of those authors!) Please check all references thoroughly to ensure the details are correct and for consistency of formatting.
Many thanks again for the chance to review this interesting manuscript, and best of luck with the remainder of the publication process! :-)
Author Response
A detailed response to the reviewer's comments is provided in the attached file.

Reviewer 2 Report
Re: jcm-1001373: Comments to Authors
General comments
This study represents an excellent piece of research work on an important public health concern facing aging society, multimorbilidy (MM) in older adults. The study methodology and analytic approaches were presented with strong scientific rigor. I believe this study will have general interests to broader audience from clinicians and researchers working in the field, despite the topic itself is not brand new. Meanwhile, I would like to share some specific comments and suggestions for authors to consider.
Specific comments
- Page 10, Line 86: “…grouped into 60 …” Is the “60” a typo of 6?
- Page 3, “2.5. Analytic approach” under Methods
- For the Cox proportional hazard regression on first unplanned hospitalization, authors claimed that “Models were censored for death, …”. My questions are:
- How many participants died each year, overall and for each MM pattern?
- How many participants were lost to follow-up or dropped out of study before the end of follow-up? How were these situations handled in the model?
I’d add a supplementary table to present these critical data for censoring events (N & %) according to 6 MM patterns.
- For the negative binomial (NB) model, can authors clarify a bit further on Length of Stay (LOS)? For instance, how well did the LOS fit a NB distribution than a normal one? If not, what are the reasons to choose NB over normal model? Please provide some technical details.
- Table 2 on NB model of Length of Stay:
- The column heading says “IR per 10-person years”; yet the subheading listed “Length of Stay (days)”. This caused confusion and requires some clarification.
Similarly, Table S3. Association of multimorbidity patterns with unplanned hospital care utilisation stratified by age, specified both IRs and (days) under LOS model. For instance, does the value 175.65 for Psychiatric pattern indicate number of days or number of events per 10-person years?
- If being “IRs”, then the values seemed quite large to a rare-event, discrete distribution, like NB. In addition, how could LOS have excess 0 counts? To me LOS shouldn’t apply to those never admitted to hospital, because they never come under risk for staying in hospitals.
- Figure 1. I’d add a row under Axis X demonstrating total number (%) of deaths and drop-outs under each year.
Author Response

(The authors gave the same response as above.)
